## The AgraSim (Agricultural Simulator) facility for the comprehensive experimental simulation and analysis of environmental impacts on processes in the soil-plant-atmosphere system

Joschka Neumann<sup>1</sup>, Nicolas Brüggemann<sup>2, 3</sup>, Patrick Chaumet<sup>1</sup>, Normen Hermes<sup>2</sup>, Jan Huwer<sup>1</sup>, Peter Kirchner<sup>1</sup>, Werner Lesmeister<sup>1</sup>, Wilhelm August Mertens<sup>1</sup>, Thomas Pütz<sup>2</sup>, Jörg Wolters<sup>1</sup>, Harry Vereecken<sup>2,3</sup>, Ghaleb Natour<sup>1,4</sup>

Joschka Neumann<sup>1</sup>, email: j.neumann@fz-juelich.de

<sup>1</sup>Institute of Technology and Engineering (ITE) | Forschungszentrum Jülich, Jülich, 52428, Germany <sup>2</sup>Institute of Bio- and Geoscience – Agrosphere (IBG-3) | Forschungszentrum Jülich, Jülich, 52428, Germany <sup>3</sup>Institute of Crop Science and Resource Conversation (INRES), University of Bonn, Bonn, 53115, Germany <sup>4</sup>RWTH Aachen University, Aachen, 52062, Germany

#### Abstract 15

The AgraSim large-scale research infrastructure is an experimental simulator consisting of six mesocosms (each mesocosm consisting of an integrated climate chamber, plant chamber and lysimeter system) for studying the effects of future climate conditions on plant physiological, biogeochemical, hydrological and atmospheric processes in agroecosystems, which was designed and built by the Forschungszentrum Jülich.

AgraSim makes it possible to simulate the environmental conditions in the mesocosms in a fully controlled manner under different weather and climate conditions ranging from tropical to boreal climate. Moreover, it provides a unique way of imposing future climate conditions which presently cannot be implemented under real-world conditions. It allows monitoring and controlling states and fluxes of a broad range of processes in the soil-plant-atmosphere system. This information can then be used to give input to process-models, to improve process descriptions and to serve as a platform for the development of a digital twin of the soil-plant-atmosphere system.

35

#### 1 Introduction

50

55

Climate change in combination with a steadily growing world population and a simultaneous decrease in agricultural land is one of the greatest global challenges facing mankind (Molotoks et al., 2020). Investigating the effects of the changes in air temperatures and precipitation that have already occurred and are still to be expected in the future on the fundamental processes in agroecosystems, which form the basis for the sustainable management of arable land while maintaining other ecosystem services, such as the recharge of clean groundwater, the storage of carbon and nutrients and the preservation of biodiversity, is of central importance here.

In this context, IBG-3 at Forschungszentrum Jülich decided to establish an "agricultural simulator" (AgraSim), which enables research into the above-mentioned effects of climate change on agricultural ecosystems and the optimization of agricultural cultivation and management strategies with the aid of combined experimental and numerical simulation. ITE is the technology and engineering partner with the task of development and construction of Agrasim.

In its combination of experimental, analytical and simulation capabilities, AgraSim is unique worldwide. By intimately integrating real-time observations and modelling of states and fluxes, a digital twin of the soil-plant-atmosphere system will be created. Understanding the effects of climate change on agricultural yields, the role of the soil for resource use efficiency and the feedback to climate-relevant parameters can be studied and quantified in a way that is unique to date. AgraSim can also be used to test the cultivation of new varieties and their performance under changing climatic conditions. This opens up new opportunities for application-oriented bioeconomic modelling, in particular for the exploration and optimization of new and sustainable cultivation methods.

A comparison of previously implemented ecotrons is described in Roy et al. (2020). In addition, there are other large chamber systems, e.g. by NASA in the USA (Wheeler, 1992), at Kyoto University, Japan (Horie et al., 1995) and at Mendel Agriculture and Forest University in the Czech Republic (Urban et al., 2001), as well as small mobile variants, such as those described in (Leadley & Drake, 1993) (Morrow & Crabb, 2000). However, none of these ecotrons compares with Agrasim in terms of complexity and the variety of its features. In addition, the six-fold design of the chambers makes statistical studies possible.

The chambers' size also allows for experiments with various agricultural crops, including tall-growing crops such as maize. The primary aim of the AgraSim research facility is to study how agricultural soils and crops will react to the changing future climate conditions, such as rising temperatures, altered precipitation patterns and increased CO<sub>2</sub> concentrations in the atmosphere, and the consequences this will have for yields, soil health and the environment. The fully controllable plant chambers allow various climate scenarios to be simulated in a targeted manner, taking into account all relevant variables: air temperature and relative humidity; atmospheric CO<sub>2</sub> concentration; light intensity and spectrum; precipitation; soil temperature with a realistic vertical profile; soil moisture; and the lower hydrological boundary conditions at the bottom of the lysimeter. Key variables of ecosystem matter exchange can also be quantified, including evapotranspiration, net ecosystem exchange of CO<sub>2</sub>, CH<sub>4</sub>, N<sub>2</sub>O, soil water balance, quantity and composition of seepage water, plant growth and performance, and quantity and quality of yield. AgraSim enables the analysis of the nutrient and water use efficiency in the soil-plant-atmosphere system, and the quantification of the feedback of agroecosystems to the atmosphere under future climatic conditions. Stable isotope analysis can be used to disentangle the net fluxes of carbon dioxide, water vapor and nitrogen gases into their component fluxes. This provides the basis for incorporating these processes into model calculations for sustainable agriculture. It is not possible to do this in its entirety in the field, but only in such a sophisticated research facility.

## **2 Performance of the system**

The technical specifications of the AgraSim research facility are as follows:

| Pos. | Description                  | Performance data                                                                                                                                                                                                                                                                                                                                                                                                                                                  |  |
|------|------------------------------|-------------------------------------------------------------------------------------------------------------------------------------------------------------------------------------------------------------------------------------------------------------------------------------------------------------------------------------------------------------------------------------------------------------------------------------------------------------------|--|
| 1.   | Air temperature              | -5 +40 °C<br>Without condensation inside the plant chamber                                                                                                                                                                                                                                                                                                                                                                                                        |  |
| 2.   | Air temperature change rate  | Maximum ±5 °C h <sup>-1</sup> at a humidity of 80 % rH                                                                                                                                                                                                                                                                                                                                                                                                            |  |
|      |                              | (A larger temperature change rate is possible at lower humidity)                                                                                                                                                                                                                                                                                                                                                                                                  |  |
| 3.   | Minimum humidity             | The minimum humidity depends on the transpiration rate of the plant. Assuming a maximum transpiration rate of 2.5 mmol m <sup>-2</sup> s <sup>-1</sup> at an air temperature of 25 °C and 5 m <sup>2</sup> leaf area as well as a maximum supply air volume flow of 1,000 l min <sup>-1</sup> of dry air, this results in a relative humidity (rH) of approx. 60 %. With lower transpiration rates, significantly lower air humidities can be set (e.g. 10 % rH). |  |
| 4.   | Maximum humidity             | 80 % rH                                                                                                                                                                                                                                                                                                                                                                                                                                                           |  |
|      |                              | No condensation occurs inside the plant chamber or the atmosphere-carrying components.                                                                                                                                                                                                                                                                                                                                                                            |  |
| 5.   | Carbon dioxide content       | ca. 385 2,000 ppm                                                                                                                                                                                                                                                                                                                                                                                                                                                 |  |
| 6.   | Air pressure                 | + 4 mbar to the environment to prevent external contamination                                                                                                                                                                                                                                                                                                                                                                                                     |  |
| 7.   | Light intensity              | 0 1,000 $\mu$ mol m <sup>-2</sup> s <sup>-1</sup> on the plant chamber floor 0 2,500 $\mu$ mol m <sup>-2</sup> s <sup>-1</sup> at a distance of 2.6 m from the floor surface                                                                                                                                                                                                                                                                                      |  |
| 8.   | Light spectrum               | Spectrum similar to daylight in the wavelength range of approx. 360-740 nm (without infrared component). Variation of the light spectrum via 8 LED channels, including a channel for the UV-A range.                                                                                                                                                                                                                                                              |  |
| 9.   | Supply air volume flow       | 50 1,000 l min <sup>-1</sup>                                                                                                                                                                                                                                                                                                                                                                                                                                      |  |
| 10.  | Supply air, lowest dew point | -40 °C                                                                                                                                                                                                                                                                                                                                                                                                                                                            |  |
| 11.  | Irrigation quantity          | 6 48 1 h <sup>-1</sup>                                                                                                                                                                                                                                                                                                                                                                                                                                            |  |
| 12.  | Soil temperature control     | -5 +30 °C                                                                                                                                                                                                                                                                                                                                                                                                                                                         |  |
| 13.  | Purity                       | All relevant surfaces in contact with the atmosphere have inert properties thanks to an appropriate choice of materials (mainly glass or an inert metal coating), (Vaittinen et al., 2013) with Silconert-2000 (Silcotek, Bad Homburg Germany).                                                                                                                                                                                                                   |  |
| 14.  | Plant chamber dimensions     | 1,600 x 1,600 x 2,700 mm (L x B x H)                                                                                                                                                                                                                                                                                                                                                                                                                              |  |
| 15.  | Weight of the plant chamber  | ca. 1,500 kg                                                                                                                                                                                                                                                                                                                                                                                                                                                      |  |
| •    | TT 1 1 1 10 11 0 1 0 1 0     |                                                                                                                                                                                                                                                                                                                                                                                                                                                                   |  |

Table 1: Technical specifications of the AgraSim research facility

Figure 1: Setup of the experimental research infrastructure AgraSim.




The core of the Agrasim research infrastructure (see Figure 1 and the following Figures on the next pages) consists of a total of six plant chambers (Figure 1, d) which can be operated independently of each other. Three chambers can be seen in Figure 1, three more are located on the opposite side of the hall. The system extends over three floors. On the ground floor the plant chambers are located (Figure 1, d), which in turn are each arranged in a climate chamber (Figure 1, c, chamber-in-chamber system). The lysimeter system (lysimeter: soil container) with one lysimeter (Figure 1, f) per plant chamber is located in the basement. The process technology (Figure 1, a), which is used to condition the supply air to the plant chamber, for example, is located on the technical platform on the first floor. The ultra-pure air system (Figure 1, e) for supplying air to the plants is located outside the hall. This produces dry and purified compressed air for the air supply to the plant chambers. The compressed air enters the process technology racks (Figure 1, a) on the technology platform, where it is regulated to a defined volume flow, tempered, humidified and mixed with CO<sub>2</sub>. This conditioned air enters the plant chambers (Figure 1, d) and supplies the plants on the lysimeter (Figure 1, f) with air. The air is led out of the plant chamber through the exhaust pipe (Figure 1, b) and the pressure inside the plant chamber is regulated to a low overpressure (400 Pa). A gas sample of the supply air and a gas sample of the exhaust air from each plant chamber are taken via a valve terminal and sequentially connected to gas analyzers for the determination of water vapor, greenhouse gases (CO<sub>2</sub>, CH<sub>4</sub>, N<sub>2</sub>O) and reactive nitrogen gases (NO, NO<sub>2</sub>, NH<sub>3</sub>, HONO). The irrigation station (Figure 1, g) in the basement is used to irrigate the surface of the soil in the lysimeters (Figure 1, f).

Figure 2: Sectional view in the top view of the experimental research infrastructure AgraSim.

Figure 3: Schematic Cross-sectional view (A-A in Figure 2) of the experimental research infrastructure AgraSim.

#### 4 Plant chamber








The plant chamber (Figure 4, a) is located inside a climate chamber (Figure 4, b) and is centered and sealed on the lysimeter (Figure 4, c).

The climate chamber (Figure 4, b) supports the temperature control of the plant chamber and prevents condensation inside and outside the plant chamber by ensuring that all external surfaces of the plant chamber, as well as all pipes and hoses, are tempered to the target temperature of the plant chamber by the climate chamber air. The plant lighting is installed in the climate chamber ceiling.

The choice of materials for the construction of the plant chamber and built-in parts is severely restricted, as many materials would change the plant chamber atmosphere through outgassing and/or cause interactions between the plant chamber atmosphere, the soil, the water for irrigation or the plant itself and other materials, which is also undesirable. Some materials, for example, release heavy metals, plasticizers, hydrocarbons or xylene from paints and epoxy adhesives (Knight, 1992) Metallic surfaces made of aluminum, copper, brass, nickel and galvanized zinc can form toxins on contact with nutrient-containing irrigation water (Knight, 1992) (Graves & Adatia, 1983, p. 103-104). Plastics contain fats, acid and vegetable oil derivatives, which are subject to degradation by microorganisms, as well as ester compounds, phthalic acids, maleic acid and phosphoric acids (Knight, 1992) (Mathur, 1974), which are also undesirable in this environment. In addition, some plastics, e.g.

**Figure 4: Plant chamber.**Source: U. Limbach | Forschungszentrum Jülich

polyvinyl chloride (PVC) and polyethylene (PE), contain the plasticizer dibutyl phthalate (DBP), which is toxic to plants at high concentrations (which has already been proven in plant chambers made of plastics) (Knight, 1992). Contact between Plexiglas surfaces and water containing calcium produces gases that are harmful to plants (Mortensen, 1982). Brush motors are another source of hydrocarbons (Knight, 1992).

#### **Applicable materials**

PTFE (polytetrafluoroethylene) and glass are materials that have been shown to have a low impact on the plant's environment (Knight, 1992). O-rings made of Viton or FKM (fluororubber) only release very small amounts of hydrocarbons when exposed to radiation (Knight, 1992). At the same time, O-rings are often concealed and are therefore not irradiated by plant lighting. Furthermore, a surface coating of stainless steel with Silconert-2000 from Silcotek (Bad Homburg, Germany) results in an inert surface (Vaittinen et al., 2013).

## Structure

- The outer surfaces of the plant chambers were realized using the following types of glass sheets from the manufacturer Schott (Mainz Germany) and the company Kastenholz (Frechen Germany):
  - Side walls: Laminated safety glass consisting of white glass ESG-H (t = 6 mm), 1.52 mm PVB Interlayer clear UV, white glass ESG-H (t = 8 mm). Mirror foil was applied to the outside of the side walls to reduce the decrease in light intensity over the height (in the lower area with a height of 1 meter, the reflectance is 80 %, above a reflectance of 97 % was selected).
  - Floor: Laminated safety glass consisting of 2 x 8 mm white ESG-H glass with 1.52 mm PVB interlayer clear UV

• Ceiling: ESG-H white glass with Conturan green anti-reflective coating from the manufacturer Schott (Mainz, Germany). This glass has a very low reflectance of 1-2 % in the 450-650 nm range. In addition, the transmission is very high at over 97 % in the 450-650 nm range (Schott, 2015, p. 9, p. 12). At shorter and longer wavelengths, the reflectance is higher and the transmission lower, see (Schott, 2015, p. 9, p. 12), which is why this must be compensated for with a higher light intensity.

The plant chamber is sealed using precured, individually manufactured FKM seals (fluororubber, company Flohreus, Veitsbronn Germany). For sealing, all outer edges of the glass panes are pressed against each other with springs with a contact pressure of 3,000 N m<sup>-1</sup>. The heat exchangers (stainless steel 1.4571) and the fan blades (stainless steel 1.4301) arranged in the plant chamber have a large exchange surface with the atmosphere. They were therefore provided with the inert coating Silconert-2000 (Silcotek, 2023) (vaporization with silicon). These measures lead to very little influence on the atmosphere from the plant chamber and the built-in parts.

#### 5 Temperature control of the plant chamber atmosphere via two cooling towers

The plant lighting can generate up to 3 kW of heat load within the plant chamber. Figure 5 shows the cooling towers used to control the temperature of the plant chamber atmosphere.

The two cooling towers (Figure 5) inside the plant chamber are used to control the temperature of the plant chamber atmosphere. In addition to controlling the temperature to the desired value, the cooling towers also ensure homogeneous mixing of the atmosphere.

Figure 5: Cooling tower inside the plant chamber.


Each of the two cooling towers consists of a fan (Figure 5, c, Pelzer Ventilatoren, Dortmund, Germany,  $\dot{V} = 1,800 \text{ m}^3 \text{ h}^{-1}$  at  $\Delta p = 140 \text{ Pa}$ , with Silconert-2000 coating), a heat exchanger arranged above it (Figure 5, b, manufacturer: Schwämmle, type: ELW 4-HP stainless steel, with Silconert-2000 coating) and a mounted air distribution tube made of borosilicate glass (Figure 5, a, Aachener Quarzglas Technologie, Aachen, Germany, and Landgraf Laborsysteme HLL GmbH, Langenhagen, Germany, Ø300 mm). The flow directions within the two cooling towers are opposite, so that a diagonal flow is created within the plant chamber via the side nozzles of the air distribution pipes. This ensures good mixing of the atmosphere over the entire height of the plant chamber. All components must have a surface temperature above the dew point of the air, including the heat exchangers for cooling. At high humidity, there is therefore only a small ΔT available for cooling the plant chamber atmosphere (e.g., at 75 % rH, the dew point is only 4.7 °C below the air temperature), to ensure that condensation is avoided. However, this low permissible ΔT makes efficient cooling difficult and costly. Therefore, a high air flow rate (approx. 1,600 m³ h⁻¹ per cooling tower) and a high cooling water flow rate (approx. 1,000 l h⁻¹ per cooling tower) are required to achieve a cooling capacity of approx. 3 kW with such a low ΔT.

At the same time, the air speed should be limited so that the plants are not exposed to too strong air flow, which is why many side nozzles (air speed max 12 m/s directly after a side nozzle) were provided in the air distribution pipes (pipe on the left: 15 side nozzles, pipe on the right: 18 side nozzles, Figure 5). Individual side nozzles can be closed to adapt the flow conditions to the plant. The drive of the fan blades (Figure 5, d) was equipped with a brushless motor and arranged outside the plant

chamber. The rotary motion is transmitted into the plant chamber by means of an inert feedthrough. The rotary feedthrough also has a desirable, minimal air purge to the outside of the plant chamber to prevent contamination of the atmosphere by the bearings or the drive itself.

Figure 6 shows the target temperature (black) and the actual temperature (red) inside the plant chamber at maximum plant lighting output over a period of approx. 3 hours. The temperature was reduced from 34.5 °C to 19.5 °C. The humidity curve (blue) is also shown (the humidity was also controlled using the profile shown).

Temperature sensor: Combined sensor for temperature and humidity

Manufacturer/Type: Rotronic/HC2A-SM

• Control accuracy in this scenario:  $\pm 0.25$  °C (see Figure 6)

■ Measurement deviation at 23 °C and ±5 °C: ± 0.10 °C




Figure 6: Measured temperature and humidity curve inside the plant chamber.

### 6 Concepts investigated for temperature control of the plant chamber atmosphere

During the development phase, three concepts for controlling the temperature of the plant chamber atmosphere and for dissipating the heat load from the plant lighting (approx. 3 kW) were investigated using CFD (computational fluid dynamics) simulations.

#### Requirements for the cooling concept

- Removal of the heat load introduced by the plant lighting to a maximum of 3 kW with a ΔT between the plant chamber air and the cooling medium of less than 4.7 °C in order to be able to implement an air humidity of 75 % rH without condensation occurring in this extreme scenario. This corresponds to a required cooling capacity of 640 W K<sup>-1</sup> ΔT.
- As space-saving as possible to minimize the impact on the growing space for the plants (diameter: 1,129 mm, height: 2,500 mm).
- Use of inert components, especially for components that have a large contact surface with the atmosphere.

The concepts examined were:





- 1. Cooling via a temperature-controlled plant chamber ceiling
- 2. Cooling via heat exchangers in the side walls of the plant chamber
- 3. Implemented and described above: cooling via two cooling towers equipped with heat exchangers and fans

## 6.1 First cooling concept: Cooling via the plant chamber ceiling

In this concept, the ceiling glass panel of the plant chamber was double-walled, with a cooling liquid flowing through the gap. This allowed the ceiling to be tempered to the desired temperature. A silicone oil was chosen as the cooling medium, which only slightly affects the light spectrum of the plant lighting, even in the UV range. In order to achieve good heat transfer on the inner surface of the plant chamber ceiling, a ceiling fan was provided. The fan blades of which are made of glass and therefore only slightly interfere with the plant lighting. The walls of the plant chamber were also cooled from the outside by the climate chamber air. However, the influence of the climate chamber air on the internal wall temperature of the glass panels was significantly lower than the influence of the tempered ceiling glass panel with plant chamber air flowing against it.

#### 280 Input variables and boundary conditions for the CFD heat balance simulation

- Ceiling fan data: Ø250 mm | Axial air speed: 10 m s<sup>-1</sup>
- Temperature of ceiling glass panel: 3 °C
- Air temperature of climate chamber: 3 °C
- The application of the heat load of 3 kW is simplified as follows:
  - o 52% of the energy is absorbed by the floor slab outside the lysimeter (corresponding to the area share).
  - o 36% of the energy is considered as heat flow on the upper sides of the leaves, Reflection of typically 15% of the radiation is neglected.
  - o 12% of the energy is transferred to the bottom of the lysimeter.


#### Result of the CFD heat balance simulation









The evaluation showed that the temperature homogeneity was not sufficient if plants are located in the chamber which will prevent even mixing of the cooled air in the whole chamber. High temperatures of up to 30 °C will occur in the lower area of the plant where the air exchange is strongly decreased by the leaves. Moreover, the leaves might suffer from high temperatures due to radiation and in-sufficient cooling. It should be mentioned here that the plant temperatures at the leave-air interfaces shown in Figure 7 do not consider any heat conduction through the leave or cooling effect due to evaporation and are therefore quite conservative. The main purpose of the modelled plant is to study the influence on the temperature distribution of the air in the chamber. The average air temperature of 15 °C was also clearly too high. For these reasons, this concept was not pursued further.

Figure 7: Result of CFD simulation.

## 6.2 Second cooling concept: Cooling via heat exchangers in the plant chamber walls

In this concept, a heat sink with a fan was arranged in the side wall of the plant chamber. The significantly larger cooling surface of the aluminum plate equipped with ribs was intended to significantly increase the cooling capacity compared to the first concept. In this variant, only the performance of this cooling concept was investigated for an aspired temperature difference of 5 °C between the air in the chamber and the heat sink itself without considering the whole chamber and the influence of plants inside the chamber.

## Input variables and boundary conditions for the CFD heat balance simulation

Heat sink dimensions:

 $W = 600 \text{ mm} \mid H = 1,200 \text{ mm} \mid T = 120 \text{ mm}$ 

■ Cooling fan: Ø250 mm | ca. 1,350 m³ h⁻¹

■ Temperature heat sink: 3 °C

Air temperature: 8 °C

Figure 8: Second concept: heat sink in the side wall of the plant chamber.

#### Result of the CFD heat balance simulation

The cooling capacity for this system is approx. 180 W K<sup>-1</sup> of air temperature, or 360 W K<sup>-1</sup> if two of these systems are used. The increase in cooling surface area resulted in a significant improvement compared to the first concept, but the cooling capacity was still below the required cooling capacity of 640 W K<sup>-1</sup>. Moreover, the problem of insufficient mixing of the air within the chamber, especially if plants are located within the chamber, is not solved by this concept. For this reason, this concept was also not pursued further.

In the third concept, the cooling surface was increased once again and the distribution of the air flow in the plant chamber was also improved.

## 340 6.3 Third cooling concept: cooling via two cooling towers equipped with heat exchangers and fans

This concept corresponds to the variant described in Chapter 5 and implemented in practice, namely cooling via two cooling towers equipped with heat exchangers and fans.


## Input variables and boundary conditions for the CFD heat balance simulation

Air volume flow per fan: 1,600 m³ h⁻¹
 Diameter of the fans: 300 mm

Cooling water volume flow per cooling tower: 1 h<sup>-1</sup>

Cooling water temperature: 20 °C
 Temperature climatic chamber: 20 °C

# Figure 9: Temperature distribution within the plant chamber for the third and then actually realized cooling concept.

## 355 Result of the CFD heat balance simulation

The cooling capacity of this system is 655 W K<sup>-1</sup> of air temperature, which slightly exceeds the required cooling capacity of 640 W K<sup>-1</sup>. To

dissipate the maximum heat load of 3 kW, it would be sufficient to set an average temperature difference between the cooling water and air of  $4.6~^{\circ}$ C.

Figure 9 shows that there are limited heat zones in the vicinity of the plant, but these can also be found in plants in nature.

This cooling concept was assessed as sufficient and implemented.



## 380 7 Preparation of supply air





Dried and ultra-pure compressed air (manufacturer: HPS, Düren, Germany) is used to supply air to the plant chambers (residual particles down to 0.01 µm with a separation efficiency of 99.9999 %, dew point: -40 °C, manufacturer of the compressed air fine filters and activated carbon filters: Parker, Bielefeld, Germany). The compressor type is a water-injected oil-free screw compressor for generating oil-free compressed air (manufacturer: compare, Simmern, Germany) in a fully redundant design.

The supply air volume flow is adjusted to the requirements of the experiment and the plant per plant chamber (50 ... 1,000 l min<sup>-1</sup>). After compressed air generation and volume flow regulation, the compressed air is pre-tempered by a heat exchanger to a temperature similar to the desired plant chamber temperature.

The diagram in Figure 10 shows the calculation result for the required supply air volume flow of dry supply air (dew point -  $40 \,^{\circ}$ C) for drying the plant chamber atmosphere at the maximum transpiration rate of the plant (transpiration rate at 25  $^{\circ}$ C and 5 m<sup>2</sup> leaf area: 2.5 mmol m<sup>-2</sup>s<sup>-1</sup>, corresponds to 0.45 g H<sub>2</sub>O s<sup>-1</sup>).

Example: A volume flow of 1,000 l min<sup>-1</sup> is required to keep the humidity constant at approx. 60 % rH at 25 °C (Figure 10).

Figure 10: Required supply air volume flow to keep the humidity constant depending on the plant chamber temperature.

#### 420 8 Humidification






To humidify the supply air of the plant chamber, demineralized water (electrical conductivity  $\kappa \leq 0.1~\mu S$  cm<sup>-1</sup>, concentration of active silicic acid (SiO<sub>2</sub>): < 0.02 mg l<sup>-1</sup>, no chloride ions) is heated together with a small amount of compressed air (5 l min<sup>-1</sup>) in an evaporator (manufacturer: Institut für Chemische Verfahrenstechnik, Stuttgart, Germany) to form a water vapor-air mixture and fed into the supply air. The mixture then passes through a static impeller mixer for homogenization (supplier: Institut für Chemische Verfahrenstechnik, Stuttgart, Germany). To prevent condensation, all components from the evaporator onwards are heated by a pipe heating system (type: FG200, Wagner GmbH, Wülfrath, Germany). To enable humidity control even at temperatures below freezing, an additional heater ensures that the supply air is at least +3 °C before humidification. The humidified air is mixed with the plant chamber atmosphere inside the plant chamber and cooled there by the cooling towers to the desired temperature, taking the dew point into account. This also enables humidity control at temperatures below the freezing point up to 80 % rH.

In the test shown in Figure 11, the humidity was increased from 19.5 % to 61 % over a period of approx. 3 hours, while the temperature was reduced from 34.5 °C to 19 °C during this period.

Humidity sensor: Combined sensor for humidity and temperature

Manufacturer/Type: Rotronic/HC2A-SM

■ Measurement deviation at 23 °C and ±5 °C: ±0.8 % rH

• Control accuracy in this scenario:  $\pm 0.5 \%$  rH (see Figure 11)

Figure 11: Control accuracy of the plant chamber humidity.

#### 9 CO<sub>2</sub> addition




For CO<sub>2</sub> control within the plant chamber, CO<sub>2</sub> (purity grade 4.5 | 99.995 %) is dosed into the supply air using two mass flow controllers (type: EL-FLOW Select, Bronk-horst, Kamen, Germany), mixed with the supply air using a static impeller mixer (Institut für Chemische Verfahrenstechnik, Stuttgart, Germany) and then this mixture is fed into the plant chamber. The CO<sub>2</sub> content is measured both in the supply air and in the plant chamber (type see below) in order to regulate the CO<sub>2</sub> content in the plant chamber and to quantify the influence of the plant on the atmosphere of the plant chamber and the CO<sub>2</sub> uptake/release of the plants.

Figure 12 shows the control accuracy for CO<sub>2</sub>. An increase in the CO<sub>2</sub> content from 805 ppm to 827 ppm was specified over a period of approx. 2 hours, whereby significantly larger increases are also possible.

CO<sub>2</sub> Measuring device: Manufacturer Emerson, Langenfeld, Germany

■ Type: X-STREAM (XEPG)

• Measurement deviation  $\leq \pm 15$  ppm

■ Control accuracy: ± 1 ppm (see Figure 12)

Figure 12: Target/actual comparison of CO<sub>2</sub> control within the plant chamber.

#### 10 Air pressure control

The plant chamber is kept under a slight overpressure of 400 Pa to prevent external contamination. The plant chamber pressure must be controlled very precisely (see Figure 13 for control accuracy), as the pressure increase over the surface of the lysimeter results in an apparent additional weight on the soil scale. Weighing provides important information about the water balance in the soil. The lysimeter weight can be used to determine the evapotranspiration, the exact irrigation rate, which is a very important parameter. The measurement accuracy of this weight measurement without the influence of air pressure is ± 50 g (with a mass of the lysimeter including soil of approx. 2.8 t). A pressure fluctuation of ±100 Pa in the plant chamber would result in a weight fluctuation of ±10 kg due to the lysimeter surface area of 1 m², which would make the weight measurement and the resulting irrigation quantity unusable. For this reason, it is essential to regulate and measure the air pressure very precisely as shown in the following table:

#### Pressure measurement/control

Sensor type: Differential pressure sensor 266MST

Manufacturer: ABB, Ratingen, Germany

• Sensor measurement deviation:  $\pm 0.4$  Pa

■ Control accuracy: ±4 Pa (see Figure 13)

The remaining influence of this small pressure fluctuation ( $\pm 4$  Pa =  $\pm 400$  g fluctuation of the soil weight) is offset against the measured value of the lysimeter weight, which further increases the accuracy of the weight determination.

The diagram Figure 13 shows the measured plant chamber air pressure and the soil weight over a period of approx. 10 minutes.

#### Weight measurement

Sensor type: 3 x load cell for a load of 1000 kg each

Supplier: JR-Aquaconsol, Graz, Austria

■ Measurement deviation: ± 50 g

Measuring accuracy due to the pressure control and taking into account the differential pressure measurement between the plant chamber and the environment: ±100 g (see Figure 13)
 Soil weight [kg] and plant chamber air pressure [mbar] vs. time [seconds]

Figure 13: Lysimeter weight and air pressure inside the plant chamber.

## 11 Irrigation

The water for irrigation is produced from demineralized water, minerals and other additives according to the user's specifications in a 1000-liter tank. The water is supplied via pumps (gear pump | manufacturer Iwaki, Willich, Germany | type MDG) and a drip hose (Kärcher, Winnenden, Germany | type:  $\emptyset \frac{1}{2}$ ") based on the specified climate data and is evenly distributed over the soil surface. The irrigation components (such as the hose, the flow meter and the drip hose) are regularly flushed with air to ensure that the irrigation volume is applied as precisely as possible and to prevent standing water from freezing in the hoses at temperatures below the freezing point (at temperatures below freezing point, irrigation no longer takes place). The watering volume can be set between 6 and 48 liters evenly distributed over a time of one hour.

#### 12 Plant lighting

With the plant lighting (manufacturer: Roschwege, Greifenstein, Germany), individual spectra similar to daylight (see Figure 14 for an exemplary spectrum) can be displayed using 12 different LED types (see Table 2). The maximum brightness is 2,500  $\mu$ mol photons  $m^{-2}$  s $^{-1}$ ) at a distance of 900 mm from the plant lighting.

|     | LED-color        | Wavelength |
|-----|------------------|------------|
| 1.  | UV               | 365 nm     |
| 2.  | UV               | 400 nm     |
| 3.  | UV               | 415 nm     |
| 4.  | UV               | 430 nm     |
| 5.  | Blue             | 470 nm     |
| 6.  | Turquoise        | 505 nm     |
| 7.  | Green            | 530 nm     |
| 8.  | Red              | 660 nm     |
| 9.  | Red              | 690 nm     |
| 10. | Red              | 740 nm     |
| 11. | Cold white 5000K | -          |
| 12. | Warm white 3000K | _          |

Table 2: Wavelengths of the LEDs used.

Figure 14: Daylight spectrum compared to an exemplary spectrum of plant illumination at a light intensity of  $2,500 \mu mol photons m^{-2} s^{-1}$  at a distance of 900 mm from the LED light attachment.

#### 535 13 Lysimeter system

The lysimeter system used for Agrasim is based on the technical concept established in TERENO-SOILCan (Pütz et al., 2016). The lysimeters are stainless steel containers with a surface area of 1 m², a depth of 1.5 m and are filled with a soil monolith. To control the hydraulic properties of the lysimeter, the lower boundary condition is set to the desired values using a suction cup rake in combination with a bidirectional pump, a precision tensiometer and a high-precision weighable stainless-steel container with a volume of 100 L to collect the percolate. Water is either pumped into or out of the lower region of the lysimeter. The lysimeters are installed hanging from the cellar ceiling. The high-precision weighing system of a lysimeter consisting of three load cells (model 3510, JR-Aquaconsol, Graz, Austria) and the associated percolate container allow the exact recording of the various measured variables of the water balance equation, such as evapotranspiration, precipitation, dew, percolate and water content change of the soil. In addition, sensors for measuring soil temperature and soil moisture (tensiometer TEROS-41) and TDR-Sensors (Campbell Scientific, Santa Clara, USA), soil potential sensors (CS650, TEROS-21), temperature profile sensors (TH3-S, JR-Aquaconsol) are located at various depths in the lysimeter. The soil solution is sampled using suction cups and a control unit (JR-Aquaconsol). For chemical analysis the percolate is sampled from the downward water flow via an aliquoting unit. The soil temperature is controlled via a heat exchanger loop (JR-Aquaconsol) installed in the lower area of the soil.

#### 14 Gas sampling of lysimeters

An additional system will be used to take gas samples from six depths of the soil from each lysimeter for isotope analysis with a gas analyzer from the company Picarro (Logan, USA). For this purpose, six microporous hoses (outer diameter: 9 mm) are inserted horizontally at different depths (5, 10, 20, 40, 80, 120 cm from the lysimeter upper edge) into the lysimeter through boreholes. Synthetic air (0.5 l min<sup>-1</sup>) is passed through the microporous hoses as a transport gas. Gaseous substances from the soil can pass through the shell of the microporous hoses and are transferred to the transport gas in the hoses. A total of six lysimeters are sampled in this way, which means that with six samples per lysimeter, a total of 36 gas samples are sequentially switched to the gas analyzer via valves and hose lines. From the outlet of the microporous hose at the lysimeter, the entire hose and valve section must be heated to prevent condensation on the gas-carrying surfaces. In addition, the humidity of the sample can be regulated to the optimum humidity value for the gas analyzer by adding a specific amount of synthetic air before transferring it to the gas analyzer.

The boreholes for the microporous hoses must run precisely through the ground. A specially developed drilling rig is used for this purpose (see Figure 15). It can be aligned and fixed at the lysimeter shell with the aid of cylindrical mounts in the start and target boreholes. With this drilling rig, a drill pipe is pressed into the soil, in which an auger runs and protrudes approx. 20 mm at the end of the drill pipe. The auger loosens the soil in front of the drill pipe and partially removes it so that the drill pipe can be pressed into the ground with less force and the impact on the surrounding soil is small. However, only a fine-grained part of the soil can be removed with the auger, which leads to an unavoidable slight compaction of the soil. Once the borehole has been drilled, the microporous tube is pulled in backwards using the drill pipe.

Figure 15: Drilling rig to create the boreholes for the microporous hoses and to pull the microporous hoses into the soil.

## 15 Gas sampling plant chambers








For the scientific gas analysis of the atmosphere of the plant chambers, one gas sample is taken from the supply air and one gas sample from the exhaust air from each of the six plant chambers. The gas samples are transported at a negative relative pressure of -40 kPa in order to avoid condensation within the transport hoses. The negative pressure must be maintained in all twelve transport lines at all times, which is continuously assured by one of two vacuum pumps. Each of these twelve gas samples is switched to isotope gas analysers via 3/2-way valves and analysed. A second vacuum pump also ensures that the negative pressure is maintained for the gas sample which is switched on the gas measuring device. A filter and an orifice (orifice diameter: 10 µm, heated) are located at the inlet of each transport tube to generate the negative pressure. A volume flow of 0.5 l min<sup>-1</sup> (based on 20 °C and 1,013 mbar) is required per transport hose in order to generate a negative pressure

Figure 16: Valve terminal for gas sampling out of the Plant chambers.

difference of -40 kPa via this orifice. In addition, reference gases can be switched per 2/2-way valves to the gas measuring device for calibration. Figure 16 shows the corresponding valve terminal.

## 620 16 Evaluation of the control quality and measurement accuracy of an experiment with forage peas

To evaluate the performance of the control system with regard to environmental parameters such as temperature, humidity, etc., the relevant data was analyzed in an experiment with forage peas (*Pisum sativum*).

The experiment represented a realistic case. The climate defaults used were based on a real recorded climate dataset, simulating dynamic, practice-relevant fluctuations in light, temperature, humidity, etc. with a time resolution of 1 min. In addition, the influence of rapid changes in environmental conditions (e.g. sudden changes in light intensity) on the control strategy is documented, particularly with regard to its reaction time. The focus is on the differences between actual and setpoint values, and their evaluation in the context of the system's operating strategy.

#### Parameters of the experiment








- Operating mode: The system was in regular automatic mode.
- Plant population: 120 plants of the species *Pisum sativum* (forage peas) were cultivated on each lysimeter with an area of 1 m<sup>2</sup>.
- Plant development: At the time of this evaluation, the plants had an average height of 41 cm and a Leaf Area Index (LAI) of 1.965.

#### Period of review for this evaluation:

Start: 28.05.2025 00:00
 End: 04.06.2025 23:59

Duration: 7 days

#### Climate control:

- The basis was a real, measured climate dataset from 2017 (see the respective curves of the target values in the following diagrams).
- o New setpoint values for the relevant climate parameters were specified for the system every minute.
- O The control of the CO<sub>2</sub> concentration of the chamber air was switched off, as the diurnal fluctuating CO<sub>2</sub> concentration of the outside air was used in this experiment.

## 16.1 Evaluation of the light intensity over time


Figure 17 shows the progression of photosynthetically active radiation (PAR) measured by two PAR sensors at different heights (1 m and 1.6 m above the soil surface). The light spectrum was kept constant over the entire duration of the test and is shown in Figure 18.

Figure 17: Intensity of photosynthetically active radiation (PAR) over the 7-day experimental period, measured at two different heights in the plant chamber (1 m and 1.6 m above the soil surface).

Figure 18: Comparison of the light spectrum of the plant lighting and the measured daylight spectrum.

## 670 **16.2** Evaluation of the plant chamber temperature

Figure 19 shows the course of the target and actual temperature in the plant chamber over the experiment duration of 7 days.

Figure 19: Temperature curve of the air temperature in the plant chamber over a 7-day period.

## 675 Analysis of the temperature deviation between actual and setpoint value


Table 3 shows the percentage distribution of deviations between the measured (actual) and the specified (target) air temperature over the test period (7 days).

| Criterion (Deviation) | 

Figure 20: Relative humidity and air flow in the plant chamber over a 7-day period.

#### 695 Analysis of the deviation between target and actual relative humidity



Table 4 shows the percentage distribution of deviations between the specified (target) and the actual relative humidity over the seven-day test period.

| Criterion (Deviation) | 

Figure 21: Comparison of the setpoint and the measured value of the irrigation

Figure 22: Detailed view: Comparison of the setpoint and the measured value of the irrigation

## Analysis of the deviation between target and measured irrigation quantity

Target irrigation quantity: 2.12 LiterMeasured irrigation quantity: 2.121 Liter



• Deviation: 0.001 Liter or 0.047 %

Once irrigation is complete, compressed air is used to flush the tubing behind the flow meter to ensure that the entire volume of water is transported to the lysimeter. The evaluation of the lysimeter weight in the following chapter shows that a corresponding increase in weight was recorded at the time of irrigation, which confirms the successful application of water.

## 16.5 Analysis of the lysimeter weight



Figure 23 shows the development of the lysimeter weight over time. At the time of irrigation (29.05.2025, 19:26), a clear increase in weight can be seen, which confirms the successful input of water into the lysimeter. This is followed by a continuous decrease in weight, which can be attributed to evapotranspiration – i.e. soil evaporation and plant transpiration. A pronounced day-night dynamic was observed: The decrease in weight was significantly greater during the day than the small gain at night. This effect correlates with the varying light intensity and higher temperatures during the day (see Figure 17 and Figure 19).

Figure 23: Lysimeter weight over the 7-day experiment period.

The weight change of the lysimeter corresponded well with the applied amount of water (Figure 24).

Figure 24: Comparison of the weight increase of the lysimeter at the time of irrigation the measured amount of water applied.

| Applied water quantity (measured value)                                            | 2.121 liter (see Figure 24)                     |
|------------------------------------------------------------------------------------|-------------------------------------------------|
| Weight gain of the lysimeter                                                       | 2.186 kg (see Figure 24)                        |
| Deviation between lysimeter weight measurement and irrigation quantity measurement | 0.065 kg<br>3.06 % related to irrigation volume |

## 16.6 Soil temperature




Figure 25: Soil temperature at different depths over the 7-day experimental period.

Figure 25 shows the soil temperature readings of the tensiometers (TM) at different soil depths. The following observations can be derived from the data:

- The measured values of the two tensiometers close to the surface (TM10 and TM20, i.e., at depths of 10 cm and 20 cm, respectively) showed pronounced daily fluctuations that correlate with the air temperature. These fluctuations decreased significantly with increasing soil depth.
- The soil temperature at the lower end of the lysimeter was actively maintained constant at 13 °C by means of a heat exchanger loop in the soil. Accordingly, both the deepest tensiometer (at a depth of 120 cm) and the PT100 temperature sensor at a depth of 140 cm measure almost constant temperatures of around 13 °C in this area.
- Overall, there was a vertical temperature gradient within the lysimeter: while the upper soil layers were strongly influenced by the fluctuating air temperature, the heat exchanger loop at the bottom of the lysimeter ensured a stable thermal boundary condition in the lower area.

#### 17 Control system of AgraSim





Programmable logic controllers (PLCs) from the Siemens SIMATIC S7-1500 family are used to control AgraSim. Each of the six AgraSim systems has its own CPU, which ensures the independence of the individual chambers. Furthermore, an additional PLC controls and monitors all infrastructure systems, such as the central gas supply, the central chillers, the compressed air system and the demineralized water treatment (fully desalinated water). The individual chambers can obtain the necessary information from the central control system via a facility-wide IP network and communicate with other control systems in the facility. For example, data is also exchanged with the climate chamber control system, the plant lighting and the lysimeter system. Remote access via VPN server (Virtual Private Network) was implemented via the plant network and a GSM router (Global System for Mobile Communications), which allows various manufacturers of components (e.g. the climate chambers) to gain access to their control system for service purposes without having access to the control systems of other manufacturers. Where possible, the process sensors and actuators are connected to the control system in a decentral manner. This reduces the cabling effort to a minimum and the available installation space can be optimally utilized. By using the IO system of the Siemens SIMATIC ET200SP family, the desired modularity and flexibility of the control system was achieved. Various bus systems are used for communication with the sensors/actuators, e.g., Profinet, Profibus, RS232, Modbus RTU and SDI-12 bus. The system is visualized, fully operated and monitored via a central master computer in a control room.

Figure 26: Setup of the network for the entire AgraSim system.

The plant chamber atmosphere follows the previously defined default values, such as air temperature, humidity, etc. from climate data profiles. The climate data profiles are based on measurements taken in the field or self-generated data profiles. These profiles cover at least a full year. FTP (File Transfer Protocol) is used as the data protocol in the facility's network to provide the climate data profiles and store all measured values. A separate FTP-server is therefore installed in the network. The control system uses the specified climate data to calculate the set values for the corresponding actuators for precise control of the environmental conditions.

As SCADA system (Supervisory Control and Data Acquisition) Siemens SIMATIC WinCC Professional is used. Visualization was implemented using multi-monitor operation (four 32" monitors). Other SCADA functions include user administration with different operating rights, language switching, event recording, a notification function and measurement data acquisition. Using the Alarm Control Center from Alarm IT Factory GmbH, the events from the SCADA system are transferred to an app for mobile devices. In this way, the system informs the operator about the status of the facility while he is not present.

#### 18 Assembly of the experiment

A prototype was initially built, on which the overall concept, all controllers, sensors and measuring devices were tested. Over 50.000 purchased and manufactured parts were processed for the further construction of the entire system AgraSIM. All components were managed in a database, the order and production statuses were documented and the components were finally sorted and commissioned according to assembly and component ID. The CAD model of the entire system served as the basis for assembly.

#### 19 Maintenance









It is foreseen to run experiments over several years, including growth and dormant period of the plants. The Maintenance work is specifically scheduled for a period outside the growth period, e.g. after crop harvest. The maintenance work is bundled in order to keep downtimes of the facility as short as possible.

#### Breakdowns:

- In general, the persons responsible for the system are notified at an early stage via a cell phone app in the event of slightly deviating control values, i.e. before a system failure, in order to prevent a system failure as far as possible.
- In the event of a system failure, the maintenance personnel are informed immediately via a cell phone app to enable rapid recovery.
- In the event of a system failure, the impact on the experiment, depending on the length of the system failure, must be checked individually.
- The same experiment can be run several times in parallel in different plant chambers.

#### 20 Summary








The AgraSim large-scale research infrastructure is an experimental simulator consisting of six mesocosms (each mesocosm consisting of an integrated climate chamber-plant chamber and lysimeter system) for studying the effects of future climate conditions on plant physiological, biogeochemical, hydrological and atmospheric processes in agroecosystems, which was designed and built by the Forschungszentrum Jülich.

AgraSim makes it possible to simulate the environmental conditions in the mesocosms in a fully controlled manner under different weather and climate conditions ranging from tropical to boreal climate. Moreover, it provides a unique way of imposing future climate conditions which presently cannot be implemented under real-world conditions. It allows monitoring and controlling states and fluxes of a broad range of processes in the soil-plant-atmosphere system. This information can then be used to give input to process models, to improve process descriptions and to serve as a platform for the development of a digital twin of the soil-plant-atmosphere system.

Each mesocosm consists of a temperature-controlled and weighable soil lysimeter unit with intact soil columns (1 m² surface area and 1.5 m depth) and a transparent, fully controllable plant chamber within a temperature-controlled climate chamber with an LED light source that can provide light in the wavelength range of 360-740 nm very similar to the natural solar spectrum with a maximum intensity of 2,500 µmol of photosynthetically active photons m⁻² s⁻¹. With an in-house developed, fully automated process control system, defined climatic and weather conditions as well as air compositions can be set and either kept constant over longer periods of time or varied on the basis of a predefined weather data profile. The implementation involves a considerable amount of in-house development by the Central Institute of Engineering, Electronics and Analytics (ITE) in close cooperation with the Institute of Bio- and Geosciences – Agrosphere (IBG-3) at Forschungszentrum Jülich as well as other institutes and external collaborators, as comparable measuring platforms and experimental simulators do not yet exist.

High demands are placed on the plant chamber and the process technology. The inner surfaces of the plant chambers have the purest and most inert properties possible, with the aim of minimizing interactions between the ambient air of the plants and the chamber wall. Strong LED-based plant lighting provides light conditions similar to daylight, which prevents too large heat input into the chamber. A new concept was developed and implemented to dissipate this heat, which is described in this publication. An important point to consider in the development was to avoid condensation at all times, as condensation dissolves gas molecules from the air in the condensate, changing the isotope composition and thus impeding the atmospheric measurements.

The tasks for the process technology to control the entire system are extensive and varied, which is why an individual, customized solution had to be developed for this purpose. These include precise control of the supply air volume flow, pressure, humidity, CO<sub>2</sub> content, air temperature, light intensity within the plant chamber, soil temperature and irrigation.

#### 21 Outlook






The AgraSim facility is currently in the start-up phase. The soil planned for the future experiments is then selected and placed in the lysimeters. The soil will be excavated with the lysimeters in order to preserve the natural soil structure. Once the lysimeters have been installed in the system, the first experiments will be started.

Each plant chamber is also already prepared for the addition of ozone gas exposure. This would also allow the test plants to be exposed to an increased ozone concentration. However, this system has not yet been implemented. It can be added at a later stage.

The concept described can also be implemented with significant modifications. For example:

- Adapted dimensions and shapes of the atmosphere chamber (plant chamber).
- Adapted air supply, e.g. by mixing the air with various pure gases, adding volatile organic compounds (VOCs), fine particles or similar.
- Adapted pressure conditions, e.g. use of a lower pressure in the chamber.
- Use of other sensor systems to examine the plant and integration of these sensors into the facility control.
- Adapted light spectrum and brightness.

By adapting the concept to the user's requirements, completely different applications are also conceivable, such as the "Cactus" system also planned by ITE and ICE-3 at Forschungszentrum Jülich. The system provides a defined atmosphere in a small chamber (Ø200 mm, length 1,500 mm) in order to calibrate gas sensors under a variety of conditions. The chamber air is composed of several pure gases, humidified, mixed with volatile organic compounds (VOCs) and fine particles and tempered. Both positive and negative pressure can be set. The chamber itself is inert and clean due to the Silconert-2000 coating. It would also be conceivable to use AI (artificial intelligence) to evaluate the test data, e.g. to analyze correlations and dependencies. AI could also be an auxiliary tool for controlling the measured variables (e.g., CO<sub>2</sub>, temperature, etc.) by using

AI to determine the optimum control parameters for each measured variable and the current scenario.

#### 895 22 Author contribution

The large-scale experiment Agrasim was developed and the construction supervised by the entire team. Project management was the responsibility of JN on the technical side and NB and TP on the scientific side. Simulations and calculations were carried out by JW. The mechanical construction was performed by JH and WM. The electrical planning was implemented by PK and PC, the design and development of the control system was the responsibility of PC. WL was significantly involved in the assembly of the experiment. NH supervises the system in experimental operation and contributed to the development and construction. As institute directors, NG and HV laid the foundations for the development, construction and operation of such a large-scale experiment. The paper was conceived and written by JN, NB, PC, NH, JH, PK, WL, WM, TP, JW, HV and GN.

#### 23 Competing interests

The contact author has declared that none of the authors has any competing interests.

#### 24 Acknowledgements

We thank the Helmholtz Association for providing funding for Agrasim within the program "Changing Earth - Sustaining our Future" of the research field Earth and Environment.

Finally, I would like to thank the main workshop of ITE for their technical support and professional service. Thanks to the main workshop. All equipment, materials and components were provided and installed on time and in perfect condition.

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
