# Peer review of "The AgraSim (Agricultural Simulator) facility for the comprehensive experimental simulation and analysis of environmental impacts on processes in the soil-plant-atmosphere system"

_EGUsphere, 2024_

## Referee Comment (RC1)

**Summary**: The paper details the design and specification of a large-scale simulator facility, its capabilities, and its potential to simulate environmental changes and pollutant impacts on plants. It outlines a promising foundation for future studies on climate change impacts on plant adaptation and pollution effects, creating opportunities for expanded experimentation in a controlled environment.

**Major remarks and suggestions**:
1. The introduction should explicitly state the primary goal and research gap to help understand the novelty the facility's development brings to the research domain. To help, we suggest including a comparison table to highlight how this system differs from or improves upon similar global facilities, particularly regarding key parameters such as pollutant control, temperature regulation, etc.
2. For better readability, a performance summary table with statistical data for key parameters achieved versus technical specifications (e.g., temperature control, and water exchange) could be included.
3. Could you outline a typical experimental plan (system's performance under various conditions) aimed to test this facility, including the duration of the trial, expected results, desired efficiency and regulations?
4. A discussion on the potential extended scope of testing to include extreme climatic events would be of interest. In a configuration test case like this, which measured parameters are typically used to study long-term plant adaptation under prolonged exposure to simulated climate changes or pollution?
5. Could you be precise on how the regulation system would respond to changes in the root environment, particularly in terms of adapting and handling heat management effectively?
6. Could you be precise on how the system manages pollutants and drainage water in the current large-scale testing facilities?

**Minor remarks and suggestions**:
1. The title could be made clearer to emphasize the novelty of the AgraSim simulator. If this is the first time AgraSim is being introduced, including terms like "innovation", "development" or similar would highlight its significance.
2. Ensure *all* figures are properly labelled and are self-explanatory. For example, in Figure 1, labels (a-g) should be clarified in the captions, with brief names included in the figures where possible. In Figures 4 and 9, it would be helpful to explain what the axes represent. Additionally, Figure 6 is not currently referenced in the text and should be done appropriately. Figure 8 is labelled twice—once before Figure 9 and once after. It would be helpful to verify the correct order.
3. For better readability, the Plant Chamber section could be simplified by summarizing key restricted and/or applicable materials in a table format which can include the material names, their corresponding specifications and references.
4. What is the average duration of an experiment?
5. How do you manage the failure of an air conditioning channel?

---

## Author Response (AR1)

**General comments**

This paper is well-constructed and well-articulated.

However, to enhance the overall description of such equipment, a synthetic presentation of the extreme climates covered and simplified schematic illustration of the whole system? would be appreciated.

- Two new schematic illustrations of the whole system were added on page 5.

- The limits of the system, e.g. maximum and minimum temperature, maximum rate of temperature change, etc. are listed on page 3. From this it can be derived which extreme climate scenarios can be mapped with this system.

- The broad range of climatic conditions that can be simulated with the AgraSim facility are also mentioned in the Abstract and on line 20: "AgraSim makes it possible to simulate the environmental conditions in the mesocosms in a fully controlled manner under 20 different weather and climate conditions ranging from tropical to boreal climate".

The key scientific and technical locks addressed in this paper could be more highlighted. Finally, a summary of the performances achieved after the first tests compared to those expected would be welcome.

- A new chapter has been added (Chapter 16, from p. 21 – p. 27) in which the precision and deviation of the control of the system in an experiment under real conditions are described.

Please do find below remarks and suggestions:

Adding a summary table with performance objectives versus reached ones would be of interest for the reader.

- The desired performance of the system was specified in detail at the start of planning. As part of the design, all components were determined in such a way that the specification, including safety factors, was met. The final tests after the construction of the system confirmed that the performance values of the system required in the specification (e.g. max. and min. temperature, temperature change rate, etc.) were met, which are sufficient for the planned experiments.

Regulation test results presented seems not to have been carried out with an external load (temperature, water ingress, pollutant gas episode, …) added in measurement cell.

- In the measurement results shown in the old version of the paper, the external disturbance variable was the plant lighting as a heat source, but there was no plant as a source of disturbance, e.g. for water vapor emission. A new chapter has been added (Chapter 16, from p. 21 – p. 27), in which the performance data and the control accuracy of the system with an existing plant population are described in an ongoing experiment.

Although, the time constant of natural event are low, one would be interested in extreme event impact on long term plant growing and soil evolution.

- Extreme weather conditions have not yet been tested in the facility with an existing plant population.
- The climate data set used to date is attached in the appendix ("climate data set Agrasim.xlsx).

Introduction should explicitly state the primary goal and research gap to help understand the novelty the facility's development brings to the research domain. For instance, by better introducing the limitation of existing outdoor experiments possibilities and limitations, for instance by acting directly on soil pollution composition.

- We have added the following paragraph to the introduction (line 66-78):
  "The primary aim of the AgraSim research facility is to study how agricultural soils and crops will react to the changing future climate conditions, such as rising temperatures, altered precipitation patterns and increased $CO_2$ concentrations in the atmosphere, and the consequences this will have for yields, soil health and the environment. The fully controllable plant chambers allow various climate scenarios to be simulated in a targeted manner, taking into account all relevant variables: air temperature and relative humidity; atmospheric $CO_2$ concentration; light intensity and spectrum; precipitation; soil temperature with a realistic vertical profile; soil moisture; and the lower hydrological boundary conditions at the bottom of the lysimeter. Key variables of ecosystem matter exchange can also be quantified, including evapotranspiration, net ecosystem exchange of $CO_2$, $CH_4$, $N_2O$, soil water balance, quantity and composition of seepage water, plant growth and performance, and quantity and quality of yield. AgraSim enables the analysis of the nutrient and water use efficiency in the soil-plant-atmosphere system, and the quantification of the feedback of agroecosystems to the atmosphere under future climatic conditions. Stable isotope analysis can be used to disentangle the net fluxes of carbon dioxide, water vapor and nitrogen gases into their component fluxes. This provides the basis for incorporating these processes into model calculations for sustainable agriculture. It is not possible to do this in its entirety in the field, but only in such a sophisticated research facility."

Is global instrumentation at air level is enough sensitive to allow

- We were not sure what the reviewer wanted to ask here because the question was incomplete.

What is the typical duration of an experiment and how breakdowns or maintenance operations could be addressed?

- Experiment duration: It is foreseen to run experiments over several years, including growth and dormant period of the plants.
- Maintenance work is specifically scheduled for a period outside the growth period, e.g. after crop harvest. The maintenance work is bundled in order to keep downtimes of the facility as short as possible.
- Breakdowns: The following information has been added to an additional chapter 19 (line 801)

  - In general, the persons responsible for the system are notified at an early stage via a cell phone app in the event of slightly deviating control values, i.e. before a system failure, in order to prevent a system failure as far as possible.

  - In the event of a system failure, the maintenance personnel are informed immediately via a cell phone app to enable rapid recovery.

  - In the event of a system failure, the impact on the experiment, depending on the length of the system failure, must be checked individually.

  - The same experiment can be run several times in parallel in different plant chambers.

Could you introduce some complementary information on how your regulation system would respond to changes in the root environment, to the simulation of heat or cold wave in terms of adapting and handling heat and mass transfer management, strong rain event, air pollution puff?

- The system is able to change the ambient conditions (air temperature, humidity, $CO_2$ content, etc.) as in nature within the values specified on p. 3. The soil temperature at the bottom of the lysimeter can also be controlled via a heat exchanger loop. The heat transfer in the soil corresponds to the natural heat transfer, as the lysimeter is strongly thermally insulated to the outside. The lower soil temperature is defined by the heat exchanger loop, while the upper soil temperature is controlled by the air temperature and radiation balance.
- The plant chamber itself must also be maintained at the target air temperature. This is possible within the limits specified on p. 3.

Could you detailed particular adaptation you have imagined to manage pollutants and drainage water in your large-scale testing facilities?

- Drainage water: The percolating water from the lysimeters is collected in containers, sampled, scientifically analyzed and disposed of properly.

- We were not sure what other pollutants are meant by reviewer.

Looking at the whole system, is energy harvesting and/or energy optimisation are integrated in the scheduling of your experiments?

- State-of-the-art technology is used for energy-intensive hardware, e.g. energy-efficient LED plant lighting and energy-efficient air treatment and cooling. Energy recovery has not been implemented.

How do you planned to manage TDR probe calibration in front of soil composition against its potential evolution in order to maintain accurate measurements?

- The TDR sensors have been calibrated for the soil used. In order not to disturb the soil structure, they should not be removed to calibrated regularly. The deviations due to a lack of regular calibration are estimated to be low, as the soil texture barely changes (except for the upper area, which is dug over once for each experiment).

Line 351, reference to fig 8 failed in the text

- Corrected.

<https://egusphere.copernicus.org/preprints/2024/egusphere-2024-1598/#discussion>